# Hydroxymethylation-Modified Lignin and Its Effectiveness as a Filler in Rubber Composites

**Nor Anizah Mohamad Aini [1], Nadras Othman [1,\*], M. Hazwan Hussin [2]**  **, Kannika Sahakaro [3]** **and Nabil Hayeemasae [3]**

[1]  School of Materials and Mineral Resources Engineering, Engineering Campus, Universiti Sains Malaysia, Nibong Tebal 14300, Malaysia; noranizah.ma@gmail.com
[2]  School of Chemical Science, Universiti Sains Malaysia, Minden 11800, Malaysia; mhh@usm.my
[3]  Department of Rubber Technology and Polymer Science, Faculty of Science and Technology, Pattani Campus, Prince of Songkla University, Pattani 94000, Thailand; kannika.sah@psu.ac.th (K.S.); nabil.h@psu.ac.th (N.H.)
**\***  Correspondence: srnadras@usm.my; Tel.: +60-45996177

**Abstract:** Kraft lignin was modified by using hydroxymethylation to enhance the compatibility between rubber based on a blend of natural rubber/polybutadiene rubber (NR/BR) and lignin. To confirm this modification, the resultant hydroxymethylated kraft lignin (HMKL) was characterized using Fourier transform infrared (FTIR) and nuclear magnetic resonance (NMR) spectroscopy. It was then incorporated into rubber composites and compared with unmodified rubber. All rubber composites were investigated in terms of rheology, mechanical properties, aging, thermal properties, and morphology. The results show that the HMKL influenced the mechanical properties (tensile properties, hardness, and compression set) of NR/BR composites compared to unmodified lignin. Further evidence also revealed better dispersion and good interaction between the HMKL and the rubber matrix. Based on its performance in NR/BR composites, hydroxymethylated lignin can be used as a filler in the rubber industry.

**Keywords:** lignin; hydroxymethylation; bio-filler; rubber composite

## 1. Introduction

Fillers are used in rubber for many reasons. They improve the mechanical and thermal properties, and also reduce the cost and weight of the rubber product. Among commercial fillers, carbon black and silica are the most widely used in the rubber industry. Several studies have reported the substitution of these fillers with other alternative fillers, such as carbon nanotubes, cellulose, protein, starch, and clays [1–8]. Recently, fillers from renewable resources have been widely used in order to replace commercially available fillers due to their sustainability and biodegradability and have been effectively used to improve the mechanical properties of rubber composite. Compared to these renewable materials, lignin is an interesting alternative due to its abundance, low cost, and renewability, which make it potential filler for rubber. Since the 1940s, researchers have reported the preparation of lignin-based rubber composites through patents and academic papers. The reinforcing effect of lignin on rubber composites has been shown by several studies, especially when certain polar materials were used.

Lignin is an aromatic polymer which is mainly comprised of three major phenylpropanoid units, namely *p*-hydroxyphenyl (H), guaiacyl (G), and syringyl (S) units, which form a complex three-dimensional structure. Several linkage types are present in lignin, including the ether linkages (C-O-C), β-β, β-O-4, β-5, etc. [9,10]. Conventionally, lignin is a by-product of the pulp and paper manufacturing industry. A major portion of this lignin is burnt and used as fuel due to its high energy

content. Only a small percentage (about 2%) of the lignin produced in the world is converted into value-added products [11–13].

The utilization of lignin as a filler in rubber composites may reduce the dependency on using oil-based materials [14–17] as fillers. However, the disadvantages of using lignin are its poor dispersion and compatibility in non-polar rubber. With the development of new technologies, scientists have been looking to improve manufactured products containing lignin as a filler. Many approaches have been used to enhance the performance of lignin-based rubber composites, including modifying the surface of the lignin [18,19], and using hybrid technologies with carbon black [20], silica [21], montmorillonite [22,23], and layered double hydroxides [24]. Hybridization has been claimed to be an effective method to reduce the viscoelastic loss of rubber compounds [20] and improve the mechanical properties of such compounds with excellent reinforcing effect [22,24]. Subsequent studies involved the modification of lignin particles by acetylation [25,26], alkylation [27,28], etc. Current efforts to develop lignin-based fillers for rubber are focused on the chemical transformation of lignin to increase the crosslinking between lignin macromolecules with formaldehyde as a crosslinking agent. Jiang et al. [29] reported the successful utilization of hydroxymethyl lignin in styrene-butadiene rubber (SBR), improving mechanical properties, such as tensile strength and tear strength. Hydroxymethyl groups were introduced onto the C5 positions of G units in the lignin structure, thus producing further crosslinking condensation reactions between lignin macromolecules. Furthermore, hydroxymethylation modification has also been used to produce lignin nanoparticles [29,30].

To the best of our knowledge, the effect of hydroxymethylated lignin (HMKL)-filled natural rubber/polybutadiene rubber (NR/BR) composites has not been reported. Therefore, the main aim of this work is to focus on the modification of kraft lignin (KL) by using hydroxymethylation and to investigate the effectiveness of HMKL as a reinforcing filler in NR/BR composites. The structure of HMKL was investigated using Fourier transform infrared (FTIR) and nuclear magnetic resonance (NMR) spectroscopy in order to confirm the reaction between formaldehyde and lignin molecules. Additionally, the compounding formulation was also designed to incorporate carbon black (CB). This was done to ensure that hybrid fillers based on modified lignin/carbon black (HMKL/CB) would be able to provide synergistic properties. Modified lignin was used at various concentrations ranging from 5 to 20 parts per hundred rubber (phr) and was hybridized with carbon black with a total filler concentration of 50 phr. The physical, mechanical, and thermal properties of HMKL-filled NR/BR composites were further evaluated.

## 2. Materials and Methods

Natural rubber (NR), SMR 10 was supplied by the Rubber Research Institute of Malaysia (RRIM), and polybutadiene rubber (BR), BR9000 was supplied by Zarm Scientific (M) Sdn. Bhd (Penang, Malaysia). Kraft lignin (weight average molecular weight, $M_W$ = 3526 g/mol, density = 2.64) was extracted in-house from kenaf biomass and sieved to an average particle size of less than 250 μm. Sodium hydroxide (NaOH), sodium sulfide ($Na_2S$), ammonium hydroxide ($NH_4OH$), formaldehyde, acetic anhydride, pyridine, sulphuric acid ($H_2SO_4$), and hydrochloric acid (HCl) were purchased from Merck (Petaling Jaya, Malaysia). Deuterated dimethyl sulfoxide (DMSO-$d_6$) was purchased from Sigma-Aldrich (Darmstadt, Germany) and carbon black (N220) was purchased from Cabot Corporation (Alpharetta, GA, USA). Zinc oxide (ZnO), stearic acid (SA), *N-tert*-butyl-2-benzothiazylsulphonamide (TBBS), treated distillate aromatic extracted (TDAE) oil, paraffin wax, *N*-phenyl-*p*-phenylenediamine (6PPD), tetramethylthiuramdisulphide (TMTD), 2,2,4-trimethyl-1,2-dihydroquinoline (TMQ), and sulfur were supplied by Bayer (M) Ltd. (Penang, Malaysia).

The kraft lignin was modified using a hydroxymethylation method outlined by Popa et al. [30]. This modification was performed to substitute hydroxymethyl groups in the lignin structure and was expected to enhance the interaction between the lignin surface and the rubber matrix. A total of 10 g of lignin was mixed with 47 mL of distilled water in a 500 mL round-bottom flask. The mixture was stirred for 2 h at room temperature. Next, 1.29 g of a 50% NaOH solution and 3.14 g of a 25%

NH$_4$OH solution as catalyst were added to the lignin suspension. Then, the mixture was shaken for 2 h. Furthermore, 6.7 g of a 37% formaldehyde solution was introduced into the system. The whole system was placed in a water bath for further reaction at a temperature of 85 °C for 4 h. After that, the desired product was precipitated by adding 1.0 M HCl solution until the pH was 2. Finally, modified lignin was separated by centrifugation and washed three times with excessive distilled water and dried at 50 °C for 24 h. The resultant formaldehyde-modified lignin, i.e., HMKL, had a molecular weight of 4732 g/mol.

Subsequently, NR/BR/HMKL composites were prepared through laboratory-scale open two-roll mills with the formulation listed in Table 1. The NR and BR were first mixed with the lignin, carbon black, and other rubber ingredients (ZnO, SA, TDAE, 6PPD, TMQ, paraffin wax, TBBS, TMTD, and sulfur). The blend was then removed and kept at room temperature for 24 h. Finally, the sample was press-cured into a 2 mm thick sheet at a temperature of 150 °C for the optimum cure time (T$_{90}$), which was determined using an MDR 2000 moving die rheometer (Alpha Technologies, Akron, OH, USA). The rubber composites were named NR/BR/KL or NR/BR/HMKL, where KL indicates unmodified lignin. The NR/BR/KL and NR/BR/CB50 (without lignin) composites were prepared in a similar way to the NR/BR/HMKL composites.

**Table 1.** Formulations of natural rubber/polybutadiene rubber (NR/BR) and modified or unmodified lignin used in this work.

| Ingredients | Amount (phr) | | | | | |
|---|---|---|---|---|---|---|
| | NR/BR/CB50 (without Lignin) | NR/BR/KL10 | NR/BR/ HMKL-5 | NR/BR/ HMKL-10 | NR/BR/ HMKL-15 | NR/BR/ HMKL-20 |
| NR (SMR 10) [1] | 50 | 50 | 50 | 50 | 50 | 50 |
| BR (BR9000) [2] | 50 | 50 | 50 | 50 | 50 | 50 |
| Zinc Oxide (ZnO) | 5 | 5 | 5 | 5 | 5 | 5 |
| Stearic Acid | 2 | 2 | 2 | 2 | 2 | 2 |
| Carbon Black (N220) | 50 | 40 | 45 | 40 | 35 | 30 |
| Unmodified lignin [3] | - | 10 | - | - | - | - |
| HMKL [4] | - | - | 5 | 10 | 15 | 20 |
| TDAE [5] | 5 | 5 | 5 | 5 | 5 | 5 |
| 6PPD [6] | 2 | 2 | 2 | 2 | 2 | 2 |
| TMQ [7] | 1 | 1 | 1 | 1 | 1 | 1 |
| Paraffin Wax | 2.5 | 2.5 | 2.5 | 2.5 | 2.5 | 2.5 |
| TBBS [8] | 1.2 | 1.2 | 1.2 | 1.2 | 1.2 | 1.2 |
| TMTD [9] | 0.35 | 0.35 | 0.35 | 0.35 | 0.35 | 0.35 |
| Sulfur | 1.5 | 1.5 | 1.5 | 1.5 | 1.5 | 1.5 |

[1] Natural rubber, grade standard Malaysian rubber 10. [2] Polybutadiene rubber, grade BR9000 [3] Kraft lignin. [4] Lignin modified by the hydroxymethylation method. [5] Treated distillate aromatic extracted. [6] *N*-phenyl-*p*-phenylenediamine. [7] 2,2,4-trimethyl-1,2-dihydroquinoline. [8] *N*-*tert*-butyl-2-benzothiazyl sulphonamide. [9] Tetramethylthiuram disulphide.

FTIR spectra were recorded with a Thermo-Nicolet IR 200 Fourier transform infrared (FTIR) with attenuated total reflection (ATR) at a spectral resolution of ±4 cm$^{-1}$ with 32 scans from 600 to 4000 cm$^{-1}$.

$^1$H and $^{13}$C NMR spectra were recorded on a Bruker Avance-500 spectrometer. A mass of approximately 150 mg of acetylated lignin was dissolved in 0.40 mL DMSO-$d_6$. The $^{13}$C NMR spectra were acquired at a temperature of 50 °C in order to reduce the viscosity of the solution. $^1$H and $^{13}$C NMR data were processed offline using the Top Spin processing software (Bruker, Billerica, MA, USA).

The viscosities of the rubber composites were determined using a Mooney viscometer (model AC/684/FD, SPRI) and a single-bore Rosand capillary rheometer according to the American Society for Testing and Materials (ASTM) D1646-04 method. The results are reported in terms of the Mooney viscosity of the large rotor (ML (1 + 4)$_{100}$) at a temperature of 100 °C. The samples were heated for 1 min prior to testing at a temperature of 100 °C.

The filler–filler interaction of the uncured compounds was studied by strain sweep test in the range of 0.6–100% strain at 0.50 Hz and a temperature of 100 °C by a Montech D-RPA 3000, rubber process

analyzer (RPA) instrument (Werkstoffprüfmaschinen Gmbh, Buchen, Germany). The differences in the storage shear moduli (G') at low strain (0.6%) and high strain (100%) are reported.

Cure characteristics of the rubber compounds were tested using an MDR 2000 moving die rheometer (Alpha Technologies, Akron, OH, USA) according to the ASTM D2084-01 method. The rubber compounds were tested at a temperature of 150 °C for 30 min under an oscillating rotor at one degree of cure. Cure properties are reported, including scorch time ($t_{S2}$), optimum cure time ($t_{90}$), minimum torque ($M_L$), maximum torque ($M_H$), and torque difference ($M_H–M_L$).

The crosslink density was determined by soaking a compound rubber sample with dimensions of $30 \times 5 \times 2$ mm in toluene in a closed bottle for 72 h at room temperature, 23 °C. The sample was removed and quickly wiped and weighed, $W_S$ (swollen sample weight). Then, the swollen samples were dried at 60 °C for 30 min and weighed, $W_d$ (dry sample weight). The crosslink density, $V_C$, was determined by using the Flory–Rehner Equation (1):

$$V_C = \frac{-\left[\ln\left(1 - Q_p\right) + Q_p + \chi Q_{p2}\right]}{V \times \left(Q_{p1/3} - \left(\frac{Q_p}{2}\right)\right)} \tag{1}$$

where $Q_p$ is the volume fraction of rubber in the swollen gel, $\chi$ is the Flory–Huggins interaction parameter between toluene and rubber, $\chi = 0.38$ [31], $V$ is the molecular volume of toluene, and $Q_p$ is the volume fraction of swelling rubber, which was determined by Equation (2):

$$Q_p = \frac{1}{\left[1 + \left(\frac{W_S - W_d}{W_d}\right)\right] \times \frac{\rho_p}{\rho_S}} \tag{2}$$

where $\rho_p$ and $\rho_s$ represent the density of the polymer and solvent, respectively [32].

Tensile tests were performed on the composites according to the ASTM D412 method. The tensile tests were carried out using an H10KS tensometer (Hounsfield Test Equipment Co., Ltd., Croydon, UK) with a die type C dumbbell-shaped specimen and a crosshead speed of 500 mm/min.

Scanning electronic microscopy (SEM) images of lignin-filled rubber compounds were made with a Zeiss SUPRA 35 VP instrument (Carl Zeiss NTS GmbH, Oberkochen, Germany). The fracture surface was coated with a thin gold layer in order to obtain high-quality SEM images.

The hardness of the rubber compounds was measured using a Wallace Shore A durometer (Cambridge, UK), following the ASTM D2240 method.

Compression set tests were performed according to the ASTM D395 method at a temperature of 70 °C over a period of 22 h. The percentage compression set (C%) was calculated using Equation (3):

$$C\% = \frac{t_0 - t_1}{t_1 - t_S} \times 100 \tag{3}$$

where $t_0$, $t_1$, and $t_S$ represent the thickness of the original sample, the thickness of the sample after compression, and the thickness of the spacer, respectively.

The flexing resistance of the rubber composites was tested by subjecting the samples to repeat cycling of flexure using a De Mattia flexing machine. The number of cycles before failure was recorded for this measurement. The tests were performed according to the ASTM D430 method.

Heat build-up tests were carried out using a Goodrich flexometer (Ferry Industry, OH, USA) according to the procedure described in the ASTM D623 method. The frequency of loading was 30 Hz. Measurements of the temperatures of the samples were initially made by means of a thermocouple attached to the surface of the sample, and the temperature was recorded every minute for 25 min.

Aging properties of the rubber compounds were tested under thermal conditions. The influence of thermal aging on the properties of HMKL-filled NR/BR with different loading was investigated and compared to the influence of thermal aging on the same properties of unmodified lignin-filled compound. The specimens were aged at 70 °C for 72 h in a Geer aging oven (Tabai® GPHH-200, Tabai ESPEC

Co., Ltd., Osaka, Japan) according to the ASTM D573 method. After the aging, the specimens were removed and conditioned for 24 h at room temperature before being subjected to mechanical testing for 100% tensile modulus and tensile strength. The effect of aging on the rubber composites was calculated in terms of % strength retention, which was calculated by dividing the aged tensile property (A) by the original tensile property (O) and multiplying by 100.

Thermogravimetric analysis was performed using a Pyris 6 TGA Thermogravimetric Analyzer (PerkinElmer, Inc., Waltham, MA, USA). The measurements were made in a nitrogen atmosphere in the temperature range from 30 to 600 °C at a heating rate of 10 °C/min.

## 3. Results and Discussion

### 3.1. Structural Analysis

The reaction between formaldehyde and lignin in alkaline media results in substitution on the C5 position in the guaiacyl unit and on the side-chain-bearing carbonyl groups. This reaction can continue further with a condensation reaction with the hydroxyl groups, thus reducing the hydroxyl group content [33]. In this study, during the process of hydroxymethylation, formaldehyde was added to the lignin in the alkaline medium. Three possible reactions are proposed, as shown in Figure 1. The main reaction was involved with the attachment of hydroxymethyl groups on the lignin aromatic ring [34] and the substitution of side chains by aliphatic methylol groups, as shown in Figure 1a,b, respectively. Increasing the reaction temperature causes hydroxymethyl groups to react at free positions of other lignin macromolecules to form methylene bonds, as shown in Figure 1c.

**Figure 1.** Schematic illustration of the proposed reactions of formaldehyde with lignin (**a–c**) condensation reaction.

### 3.1.1. Fourier Transform Infrared (FTIR) Analysis

Figure 2 displays the FTIR spectra of KL and HMKL. The transmission band at 3400 cm$^{-1}$ is due to O–H stretching vibration, and the bands at 2927 and 2847 cm$^{-1}$ are due to C–H stretching vibration in methyl and methylene groups, respectively. The C–H bending vibration in methyl groups can be assigned to the band at ~1460 cm$^{-1}$. The stretching vibration at 1708 cm$^{-1}$ is due to the carbonyl groups, C=O conjugated with the aromatic ring. The absorption bands at 1601, 1509, 1454, and 1422 cm$^{-1}$ are attributed to skeletal vibrations of aromatic ring macromolecules, C=C and C–H. The band for C–O stretching vibration in the guaiacyl ring is identified at 1270 cm$^{-1}$, and the bands for C–O stretching vibration in the syringyl ring are identified at 1328 and 1111 cm$^{-1}$. The band at 1213 cm$^{-1}$ is due to the phenolic hydroxyl group and ether in syringyl and guaiacyl. The band at 1035 cm$^{-1}$ can be attributed to the C–O deformation in the secondary and primary alcohol or aliphatic esters [35]. The bands at 1328, 1270, 1213, and 1111 cm$^{-1}$ observed in both types of lignin indicate that both lignins contain the same functional groups but in different amounts.

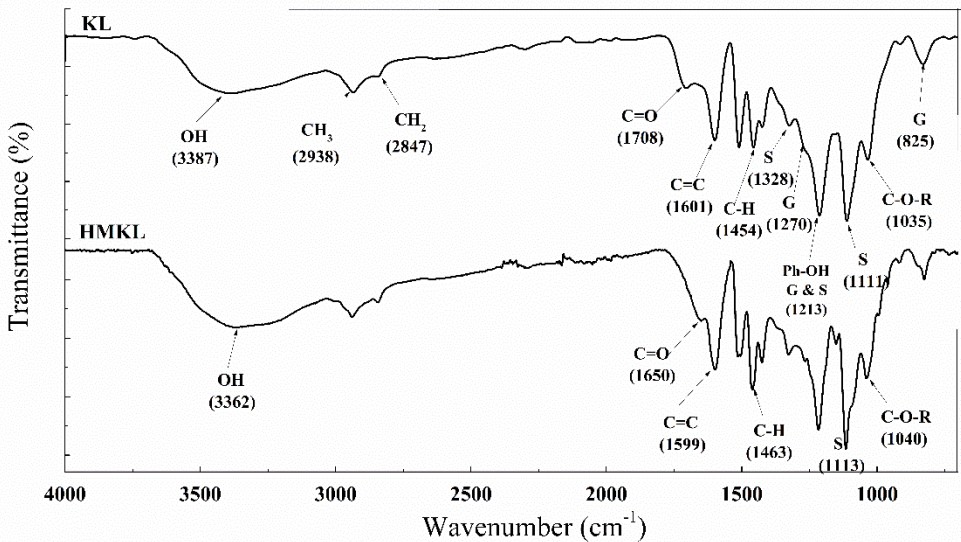

**Figure 2.** The Fourier transform infrared (FTIR) spectra of unmodified kraft lignin (KL) and hydroxymethylated kraft lignin (HMKL).

Lignin mainly contains highly reactive guaiacyl (G) moieties with a free C5 position in the aromatic ring, with small amounts of syringyl (S) and *p*-hydroxyphenyl (H) units. Therefore, the reactive G unit can actively react with formaldehyde rather than the stable S unit, which is linked to methoxy groups at the C3 and C5 positions [36,37]. In the creation of the formaldehyde-modified lignin, i.e., the HMKL, the formaldehyde reacts with the free site of G units and transforms into hydroxymethyl groups.

In the FTIR spectrum of HMKL, the band at ~3400 cm$^{-1}$, which is due to the hydroxyl group (–OH), is broader than the same band in the spectrum of KL. Intense C–H bands at 2938 and 2840 cm$^{-1}$ correspond to methoxyl groups and hydroxymethyl groups. Furthermore, the intensities of the C–H asymmetric vibration band (1463 cm$^{-1}$) and the C–O stretching vibration band for primary and secondary alcohol and ether (1040 cm$^{-1}$) are intense in the FTIR spectrum of HMKL than that of KL. On the other hand, the band at 825 cm$^{-1}$, which corresponds to the aromatic C–H out-of-plane deformation vibration of the G units at carbon positions 2, 5, and 6, is weaker in the FTIR spectrum of HMKL than that of KL. This analysis suggests the presence of hydroxymethyl groups in the lignin structure, as well as condensation resulting in the partial crosslinking of these hydroxymethyl lignin molecules. The introduction of methylol groups in the lignin structure is indicated by the increased intensity of the bands at 3400, 1463, and 1113 cm$^{-1}$ after the hydroxymethylation reaction, proving that the reaction took place within the system.

3.1.2. Nuclear Magnetic Resonance (NMR)

Figure 3 shows the [1]H NMR spectra of both KL and HMKL. The [1]H NMR spectra proves that the hydroxymethylation reaction took place between lignin and formaldehyde. The peak intensity of the spectrum of the modified lignin is observed in the range of 8–6 ppm. This is due to the substitution of hydroxymethyl groups in the aromatic nuclei of G or H units. Intense peaks in the 4.0–2.7 ppm range are related to protons in the methoxyl group. The peak related to hydroxymethylated lignin was observed to be broader than that of unmodified lignin (i.e., KL). This is due to the introduction of a functional group through the hydroxymethylation reaction. The signal from DMSO protons can be seen at 2.5 ppm. The modification of hydroxymethylated lignin can be observed in the range of 2.49–1.5 ppm, which is where the signals from acetyl groups are observed. The spectra allow the aromatic peaks to be differentiated from the aliphatic acetate peaks. These peaks correspond to the proposed reactions (a) and (b) shown in Figure 1. Proton signals, as exhibited in the range of 1.5–0.8 ppm, can be attributed to the aliphatic signal, which is related to methyl and methylene groups from the lignin macromolecules.

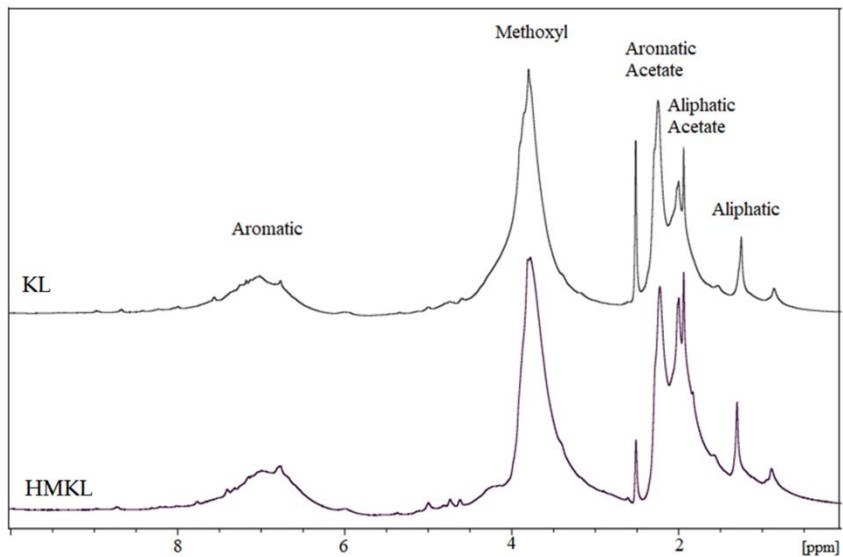

**Figure 3.** [1]H Nuclear magnetic resonance (NMR) spectra of unmodified KL and HMKL.

Figure 4 displays the [13]C NMR spectra for KL and HMKL, which are similar. The increased intensity of the peak at 154 ppm after the hydroxymethylation modification is attributed to the substitution of the hydroxymethyl functional group, which converts the G unit to a stable S unit. The peak observed at 61.3 ppm for both HMKL and KL is due to methylene or $CH_2$ resonance [38].

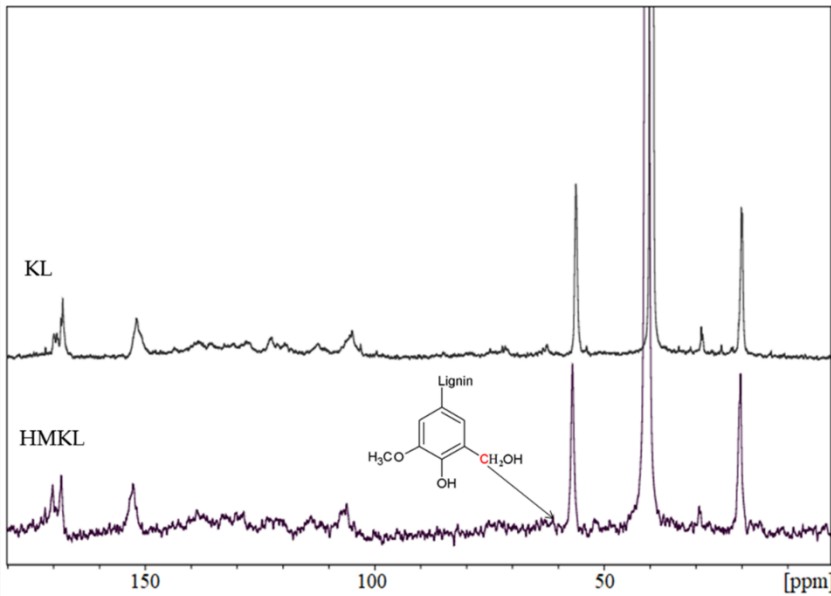

**Figure 4.** $^{13}C$ NMR spectra of unmodified KL and HMKL.

### 3.2. Mooney Viscosity and Payne Effect of Compounds

The effect of different HMKL contents on the Mooney viscosity of the NR/BR composites was investigated and compared to the Mooney viscosity of the unmodified lignin (KL10)-filled NR/BR, as displayed in Figure 5a. The highest viscosity was observed at a concentration of 5 phr of NR/BR with HMKL compared to NR/BR/KL and also NR/BR/CB50. This indicates that there is a high restriction of the mobility of the macromolecules due to the greater interaction between lignin and the rubber matrix. However, at the same lignin content, the Mooney viscosity of HMKL-filled rubber composite is higher than NR/BR/KL composite. The Mooney viscosity was found to decrease with increasing HMKL content. This is due to the higher compatibility between the filler and the rubber matrix. The reduction of the Mooney viscosity with increasing HMKL content may be attributed to the plasticizing effect of HMKL. HMKL contains low-molecular-weight lignin, which easily diffuses into the rubber chain and helps adhesion. This has a direct effect on the behavior of the rubber compound [39]. Thus, the reduction of the Mooney viscosity results in good lignin miscibility, which enhances the efficiency of the processing of rubber compounds [40–42]. Therefore, it could increase the chain mobility and improve the flow behavior of rubber compounds.

In the same way, the inclusion of lignin also affected the tensile modulus of rubber compounds, especially when high strain was applied. This phenomenon occurred since the filler networks were destroyed when high strain was applied, which suggests that the inclusion of modified lignin in the rubber matrix weakened the filler network interaction and reduced the Payne effect, as shown in Figure 5b. The Payne effect of lignin-filled rubber compounds is normally used to explain the degree of filler–filler interactions, which are mainly caused by hydrogen bonding, leading to the formation of the filler network in the rubber matrix. By decreasing the filler–filler interaction, the processability of the rubber compounds was enhanced in the present study. The other reason for the enhanced processability of the rubber compounds, i.e., the partial replacement of CB with HMKL, also plays a role in decreasing the Payne effect of the rubber compounds. However, at the same lignin content, there are no significant changes in the Payne effect of the NR/BR/KL and NR/BR/HMKL rubber composites. This is due to disruption of the well-structured filler–filler network of CB in the rubber matrix, which leads to a reduction in the strength of the filler–filler network.

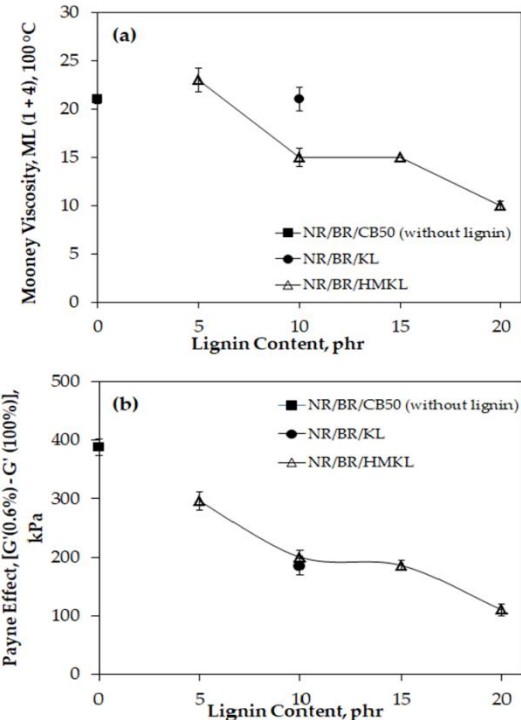

**Figure 5.** The effect of HMKL content on (**a**) the Mooney viscosity and (**b**) the Payne effect of lignin-filled natural rubber/polybutadiene rubber (NR/BR) compounds.

*3.3. Curing Characteristics and Crosslink Densities*

An analysis of the curing of NR/BR composites with KL and HMKL is illustrated in Table 2. It can be observed that the optimum curing time, $t_{90}$, and the scorch time, $t_{S2}$, of composites with HMKL are lower than those of composites with KL. This suggests that the presence of HMKL enhanced the vulcanization of the NR/BR composite due to the inclusion of the hydroxymethyl group in the lignin structure and the possibility of a reaction occurring between the hydroxymethyl functional groups of HMKL and the rubber molecules. However, the optimum curing time and scorch time of HMKL-filled NR/BR composites increased significantly with increasing HMKL content. This indicates that high levels of HMKL affect the efficiency of crosslinks in the vulcanizates and delay the vulcanization of rubber compounds [43].

**Table 2.** Curing characteristics and crosslink density of NR/BR/kraft lignin (KL) compared to NR/BR/hydroxymethylated kraft lignin (HMKL) composites.

| Sample | Optimum Cure Time, $t_{90}$ (min) | Scorch Time, $t_{S2}$ (min) | Minimum Torque, $M_L$ (dN·m) | Maximum Torque, $M_H$ (dN·m) | Torque Difference, $\Delta M$ ($M_H$–$M_L$) (dN·m) | Crosslink Density, $V_C$ ($\times 10^{-4}$ mol/m$^3$) |
|---|---|---|---|---|---|---|
| NR/BR/CB50 (without lignin) | 4.57 | 1.76 | 1.45 | 13.12 | 11.67 | 6.5 ± 0.0 |
| NR/BR/KL10 | 6.23 | 2.27 | 1.09 | 9.41 | 8.32 | 5.2 ± 0.2 |
| NR/BR/HMKL5 | 3.54 | 1.89 | 1.33 | 11.78 | 10.45 | 10.3 ± 0.1 |
| NR/BR/HMKL10 | 3.97 | 1.99 | 1.10 | 10.41 | 9.31 | 10.9 ± 0.4 |
| NR/BR/HMKL15 | 4.82 | 2.22 | 1.08 | 9.88 | 8.80 | 6.8 ± 0.1 |
| NR/BR/HMKL20 | 5.49 | 2.36 | 0.91 | 9.03 | 8.12 | 5.0 ± 0.1 |

With increasing HMKL content, the maximum torque, $M_H$, of NR/BR/HMKL displays a reducing trend, but remains higher than that of NR/BR/KL with an HMKL content of up to 15 phr. This shows that rubber compounds containing HMKL formed more crosslinks between the lignin and the rubber matrix compared to rubber compounds containing KL, which restricted the mobility of the rubber chains. Thus, the presence of HMKL enhanced the interfacial adhesion between the filler and the

rubber matrix. A similar trend is observed for minimum torque, $M_L$, which is mainly associated with the physical crosslinking between the lignin and the rubber matrix before vulcanization [44]. At low HMKL content, the rubber composite shows a high value of $M_L$, which suggests stronger physical crosslinking between the filler (CB/HMKL) and the rubber matrix. However, the trend of $M_L$ is reduced with increasing HMKL content as the lignin acts as a plasticizer in the rubber compound. It is well known that a relatively low value of $M_L$ leads to better rubber processability [45]. Furthermore, the observed $\Delta M$ indicates a reduction in crosslink density (see $V_C$ values in Table 2). All HMKL-filled NR/BR compounds exhibited a higher $\Delta M$ compared to KL-filled NR/BR compounds, except for the compound with an HMKL content of 20 phr. This is due to the higher crosslink density in HMKL-filled NR/BR compounds and could also be due to the formation of a filler–rubber network between the HMKL and rubber matrix. This will enhance the restriction of the mobility of rubber chains and significantly reinforce the strength of the rubber composite.

### 3.4. Tensile Properties

Figures 6 and 7 show the tensile properties of NR/BR composites containing KL and HMKL. For the samples with HMKL, a decreasing trend of tensile strength and tensile modulus (M100, M300, and reinforcement index, M300/M100) is observed with increasing HMKL content. However, the opposite trend was observed for elongation at break when the HMKL content was increased.

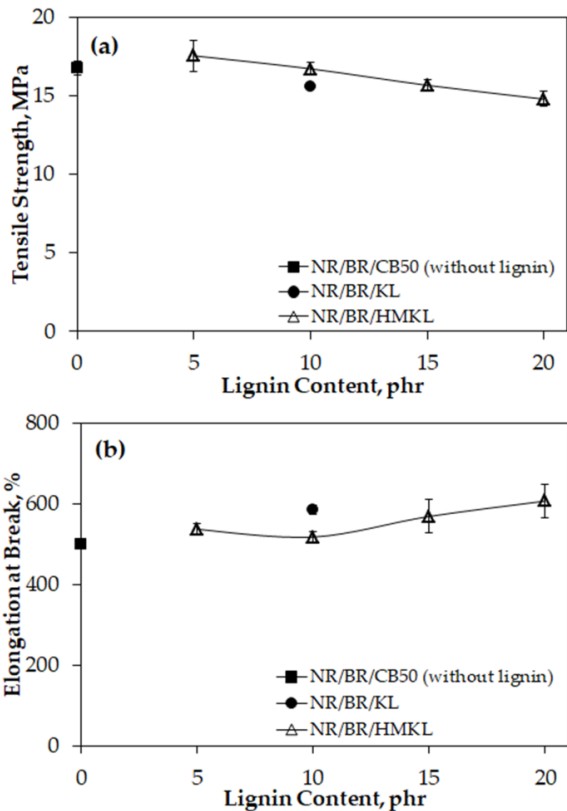

**Figure 6.** The effect of HMKL content on (**a**) tensile strength and (**b**) elongation at break, of lignin-filled NR/BR composites.

At a filler content of 10 phr, the tensile strength of the NR/BR/HMKL composite is slightly higher than that of the NR/BR/KL composite and is comparable to that of the NR/BR composite with CB50 (i.e., without lignin). This indicates that HMKL has a reinforcing effect on NR/BR and slightly enhances the rubber performance. The greater tensile strength of the NR/BR/HMKL composite is also attributed to the compatibility between the lignin and the NR/BR matrix, which can be concluded to have improved the rubber–filler interaction. As a consequence, the filler was able to support the uniform stress

transferred from the rubber molecular chains before breakage occurred [46], resulting in improved tensile strength. However, the elongation at break of the HMKL-filled NR/BR composites was lower at low HMKL contents of up to 10 phr. This is due to the rigidity of the network that formed between the HMKL and the rubber matrix, which decreased the flexibility and elasticity of the composite and reduced the deformation resistance of the rubber chains.

The tensile moduli at strains of 100% (M100) and 300% (M300) are shown in Figure 7. For both M100 and M300, the modulus decreases with increasing lignin content. However, at a lignin content of 10 phr, the modulus of the NR/BR/HMKL composite is higher than that of the NR/BR/KL composite. This could be due to the reaction between hydroxymethyl groups and the double bond of NR or BR, which could form a covalent bond. Moreover, the modulus continues to reduce with increasing lignin content at contents above 10 phr. This is due to the reduction of free volume, which leads to a decrease in the flexibility and an increase in the stiffness of the composites. This observation correlates well with the aforementioned high value of ΔM and crosslink density. Additionally, the reinforcement indexes (M300/M100) of the HMKL-filled NR/BR composites were higher than those of the KL-filled NR/BR composites at lignin contents of up to 10 phr (Figure 7c). This could be due to a reduction in the size of the lignin particles, as claimed by Jiang et al. [29] and Popa et al. [30], leading to better dispersion of lignin throughout the rubber matrix. Thus, the reinforcement index of the rubber composite for a lignin content of up to 10 phr is slightly higher than for lignin contents of 15 and 20 phr.

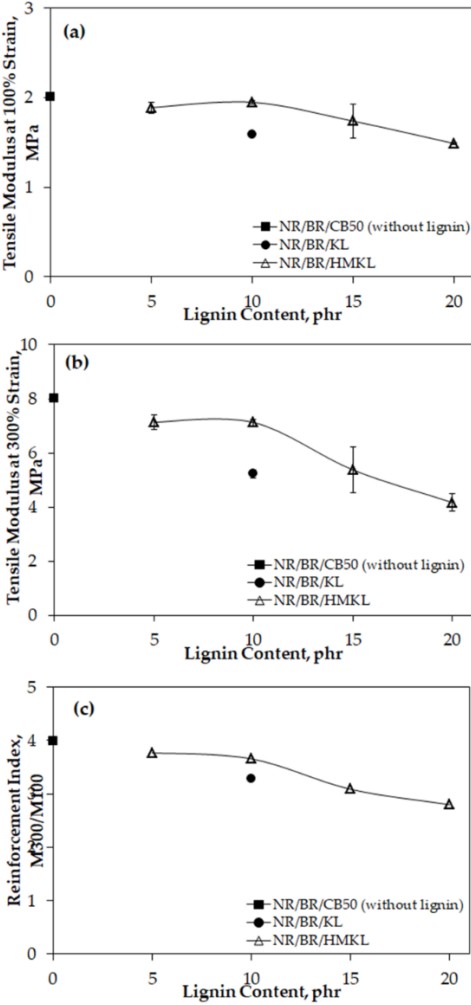

**Figure 7.** The effect of HMKL content on (**a**) the tensile modulus of lignin-filled NR/BR compounds at 100% strain, (**b**) the tensile modulus of lignin-filled NR/BR compounds at 300% strain, and (**c**) the reinforcement index of lignin-filled NR/BR compounds.

A proposed mechanism to elaborate the relationship between experimental results (e.g., tensile strength, crosslink density, and reinforcement index) with hydroxymethylated lignin content is shown in Figure 8. As depicted in the schematic illustration, the gray particles representing the HMKL formed linkages with NR/BR. Thus, the adhesion between the modified lignin and the rubber matrix is improved.

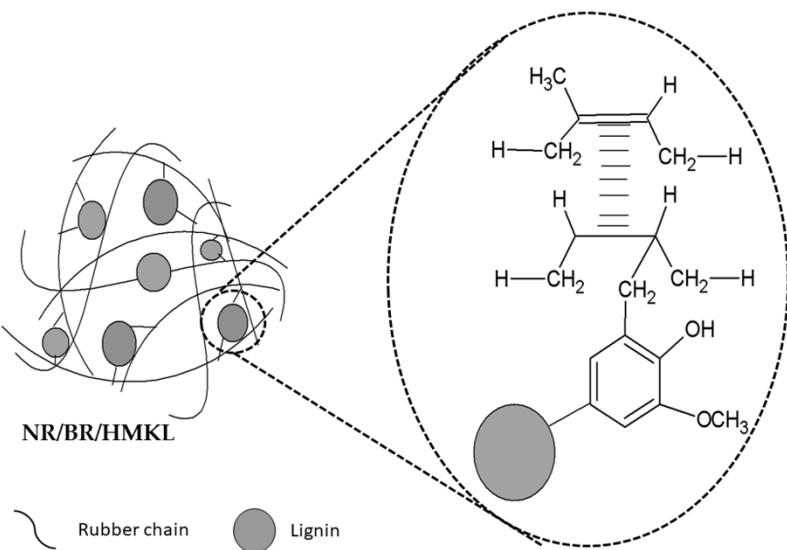

**Figure 8.** Schematic illustration of the proposed reinforcing mechanism of HMKL on NR/BR.

Filler particle size, filler dispersion, and filler–rubber interaction are three important factors that influence the reinforcement of rubber composites. The HMKL particles are uniformly dispersed in the NR/BR matrix. The HMKL particles are expected to react with the double bonds of NR or BR to form a rigid rubber network during the mixing and curing processes [29]. Under stress, the formation of covalent bonds between the NR/BR and HMKL will restrict the slippage of rubber molecules, which leads to the ultimate maximal stress at smaller deformation. However, at higher HMKL content (more than 10 phr), the lignin particles tend to self-aggregate and form agglomerations in the NR/BR matrix due to polarity differences. Ultimately, these agglomerations cannot withstand the external stress, which deteriorates the performance of the rubber compound. Therefore, the increase in the interfacial adhesion between the NR/BR and HMKL at low HMKL contents of up to 10 phr is beneficial for the transfer of external stress from the rubber matrix to the HMKL particles and results in a better reinforcing effect of the NR/BR composite.

*3.5. Compatibility of Lignin-Filled NR/BR*

The SEM images obtained from tension-fractured surfaces of the KL- and HMKL-filled NR/BR compounds are shown in Figure 9. The fractured surfaces of the KL-filled NR/BR composites (see Figure 9a) contained many holes and large amounts of filler agglomerate. This may be due to the incompatibility and poor interfacial adhesion between KL and the rubber matrix. Meanwhile, as shown in Figure 9b, the HMKL-filled NR/BR composites with an HMKL content of 10 phr had a homogeneous filler distribution due to the interaction between hydroxymethyl groups of the HMKL surface and the rubber molecules. Additionally, the morphology of the surface might have generated more shear force, leading to the break-up of the filler agglomerate. The use of more HMKL resulted in a rough surface with large holes due to the detachment of lignin from the fractured surface. This proves that the presence of low levels of HMKL improved interfacial bonding and filler dispersion.

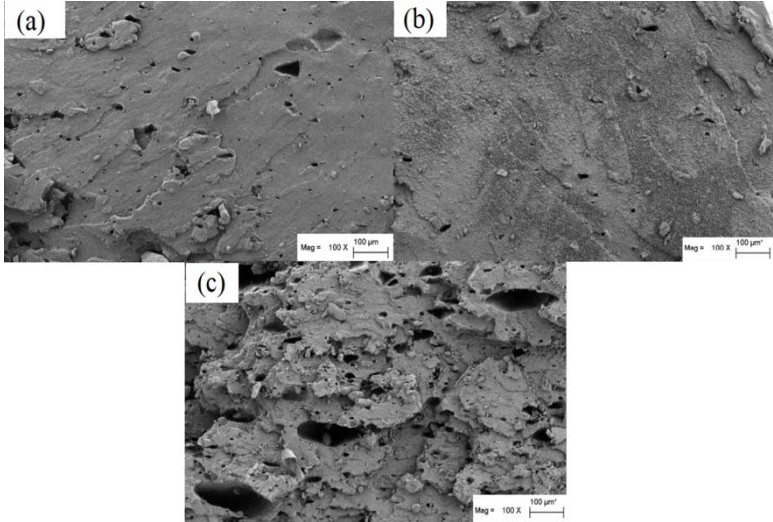

**Figure 9.** Scanning electronic microscopy (SEM) images of (**a**) NR/BR/KL10 at a lignin content of 10 phr, (**b**) NR/BR/HMKL at a lignin content of 10 phr, and (**c**) NR/BR/HMKL at a lignin content of 20 phr.

*3.6. Hardness*

The hardness of a rubber composite refers to the ability of the surface of the rubber to resist the penetration of an indenter and is related to the deformation of the rubber surface [47]. Figure 10 shows the hardness of the NR/BR composites containing HMKL for different HMKL contents. As expected, at low lignin contents of up to 10 phr, the HMKL-filled NR/BR composites have slightly higher hardness values compared to the KL-filled NR/BR composites. At the same lignin content, HMKL-filled NR/BR composites have higher stiffness compared to NR/BR/KL and NR/BR/CB50 (without lignin) composites. This is due to the formation of crosslinking between the HMKL and the rubber chains. However, a reduction in stiffness is observed with increasing lignin content, which is related to the reduction of crosslink density. This phenomenon is due to the adsorption of zinc complexes on the surface of the modified lignin, which disrupts the efficiency of sulfur vulcanization [48].

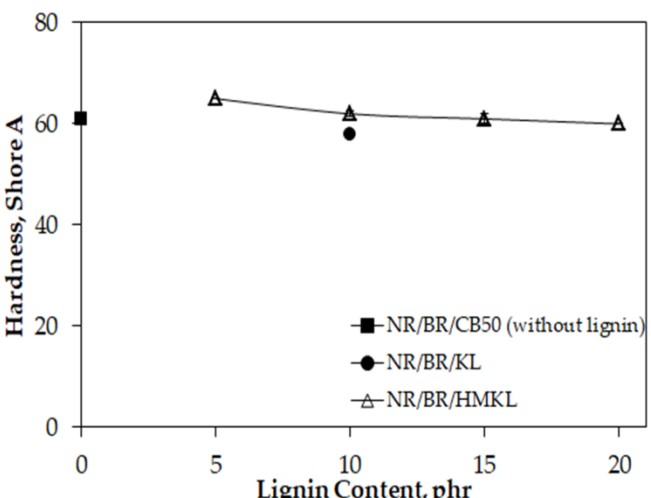

**Figure 10.** The effect of HMKL content on the hardness of lignin-filled NR/BR compounds.

*3.7. Compression Set*

Compression set tests were performed by applying stress to the NR/BR composites at certain times to determine their ability to retain their elastic properties with respect to the lignin content. The lower the compression set, the higher the ability of rubber vulcanizates to retain their elasticity.

Figure 11 displays the value of the compression set (%) for NR/BR composites containing different HMKL contents.

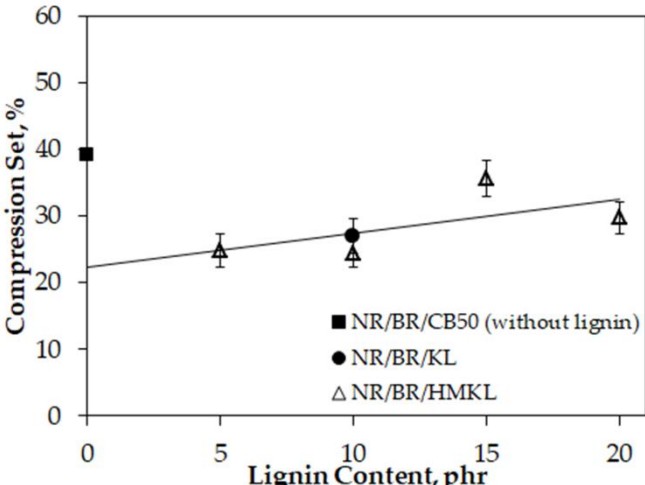

**Figure 11.** The effect of HMKL content on the compression set of lignin-filled NR/BR compounds as obtained from compression tests.

The compression set of NR/BR/HMKL composites slightly decrease at low lignin contents of up to 10 phr. This might be due to the high crosslink density, as discussed earlier [49,50]. The decrease of the compression of NR/BR/HMKL composites is also correlated to the HMKL-filled NR/BR composites, which have high values of torque, crosslink density, tensile modulus, and hardness compared to NR/BR/KL composites. This behavior is also due to the presence of hydroxymethyl functional groups in the HMKL structure, which promotes filler–rubber interaction and the formation of crosslinks, whereby it can prevent the recovery of the rubber molecular chains. However, when the HMKL content is more than 10 phr, the compression value continuously increases with increasing HMKL content. This increase is probably due to post-curing reactions which increase the tensile modulus and reducing the mobility of rubber chains.

### 3.8. Flexing Resistance

The flexing resistance of NR/BR composites for different HMKL contents is shown in Figure 12. The results show a significant decrease in the number of cycles needed for failure as the HMKL content increases. The De Mattia flexing machine is operated at a high strain level rather than at a high stress level [51,52]. Therefore, the outcomes of the flexing resistance tests are related to the tensile modulus of the rubber composites. The larger the modulus, the shorter the total flexure lifetime. Moreover, a high modulus also indicates a better per-cycle energy input. This is due to the fact that the growth of flexure cracks is an energy-dependent process resulting in poor flexure life. Furthermore, the reduction of rubber-chain flexibility increases the stiffness of rubber composites containing HMKL filler. This also results in the reduction of flexing resistance. The other assumption is the coiling of rubber chains onto the surface of the filler particles or the formation of lignin–rubber agglomerates, which result in reduced flexing resistance [53]. Thus, failure begins at the position of a lignin–rubber agglomeration, which breaks, leading to crack formation. This leads to failure when further crack propagation processes occur. However, the flexing process is very complex, since it also involves the mechano-oxidative aging process [21,54,55]. The lignin could act as an antioxidant which could partially avoid the harmful effects of flexing.

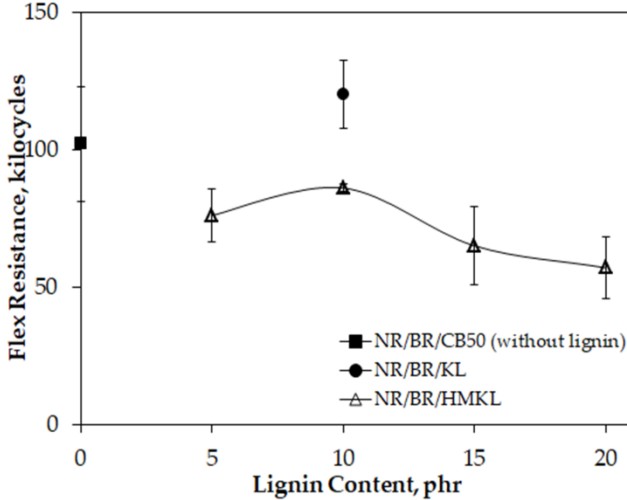

**Figure 12.** The effect of HMKL content on the flexing resistance of lignin-filled NR/BR compounds.

*3.9. Heat Build-Up*

The heat energy dissipated due to the friction between the filler particles and the rubber matrix under repeated cycles of deformation and recovery is known as heat build-up [21]. The hysteresis that generates heat build-up within rubber can lead to failure and reduce the durability of the material. Figure 13 displays the heat build-up of the NR/BR composites containing HMKL and KL.

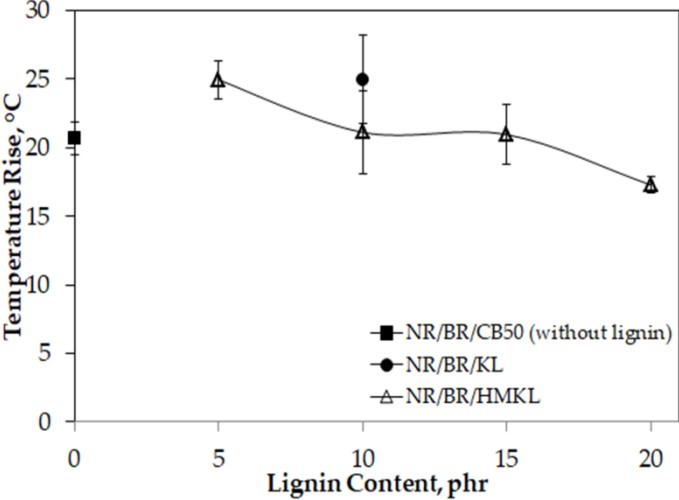

**Figure 13.** The effect of HMKL content on the heat build-up in lignin-filled NR/BR compounds.

The results show that the heat build-up reduces steadily with increasing lignin content. The heat build-up of composites with HMKL is lower than for composites with KL, except for composites with an HMKL content of 5 phr. For HMKL contents of more than 5 phr, the observed decreasing trend of heat build-up (°C) with increasing lignin content reflects good hysteresis properties. This is possibly due to the flexibility of the chain backbones of rubber composites, which correlated well with elongation at break value [48,56]. Gusev [57] reported that interfacial phenomena, rather than the filler network, were responsible for the dissipation of heat energy in rubber. This is due to the high friction between the lignin and the rubber matrix when the lignin content is increased. However, most studies of heat generation report that rubber composites with low heat build-up have low hardness [58]. Thus, the results show that as HMKL content increases, the heat build-up in NR/BR composites reduced.

*3.10. Thermal Stability*

3.10.1. Aging Resistance

The aging of NR occurs due to the impact of oxygen, heat, and stress during processing, storage, and service. The unsaturated backbone of NR is easily broken down due to environmental stresses, such as attack by oxygen molecules, especially at high temperature. This phenomenon leads to the chain scission of long molecular chains and reduces the molecular weight of rubber, which leads to less entanglement. Therefore, the ability of the rubber matrix to transfer stress is reduced, which leads to the deterioration of mechanical properties [59,60].

A variety of antioxidants were used to prevent the oxidative deterioration of the rubber composites, as well as boost their heat stability. Figure 14 displays the percentage of aging retention of NR/BR composites containing KL and HMKL. The composites were exposed to hot air conditions for 72 h, and their retention of tensile properties was then calculated. Lignin that contains many hindered phenolic hydroxyl and methoxy groups can form a special chemical structure which is able to capture free radicals and terminate the chain reaction [60–62]. Thus, it can improve the aging resistance of rubber.

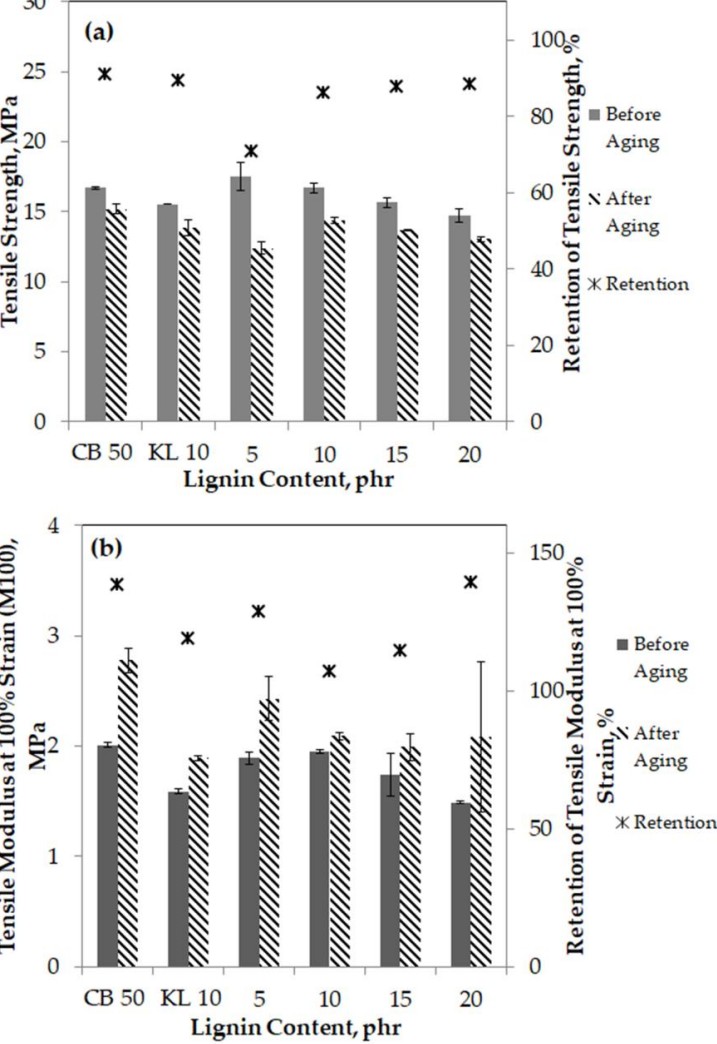

**Figure 14.** The effect of HMKL content on the retention of (**a**) tensile strength and (**b**) tensile modulus at 100% strain in lignin-filled NR/BR compounds.

As shown in Figure 14a, the tensile strengths of NR/BR/KL and HMKL-filled NR/BR with an HMKL content of 5 phr are lower than that of NR/BR with CB50. When more than 5 phr of HMKL is incorporated into the NR/BR matrix, the tensile strength increases slightly, to almost the same value as the KL-filled NR/BR composites. The percentage of tensile strength retention of HMKL-filled NR/BR composites with HMKL contents of 10–20 phr shows almost the same value as that of the KL-filled NR/BR composites. This means that less degradation occurred in HMKL-filled NR/BR composites with higher HMKL contents. It is likely that the unchanged basic structure of lignin after modification with highly branched phenolic groups prevents the rubber matrix from atmospheric degradation [44,63,64].

The increase in the retention values for M100 is shown in Figure 14b. The maximum measured value is more than 110%. This might be due to the effect of chain scission and crosslinking reactions. A macro-radical reaction occurred with the unsaturated double bond (C=C) in the rubber after hot air aging, which increased the stiffness and hardness of the rubber due to further crosslinking [65] during the post-curing process. This phenomenon might contribute to increasing brittleness, and hence, the damping of rubber composites could suffer from increased brittleness. However, as also shown in Figure 14b, the retention value of M100 is lower than that of NR/BR composites with CB50 (without lignin). Therefore, HMKL-filled NR/BR is a more stable compound during the service life under exposure to heat compared to compounds with CB50 (without lignin).

3.10.2. Thermogravimetric Analysis (TGA)

The thermal characteristics (thermogram, TG, and derivative weight loss, DTG) measured for KL-filled NR/BR and HMKL-filled NR/BR are shown in Figure 15.

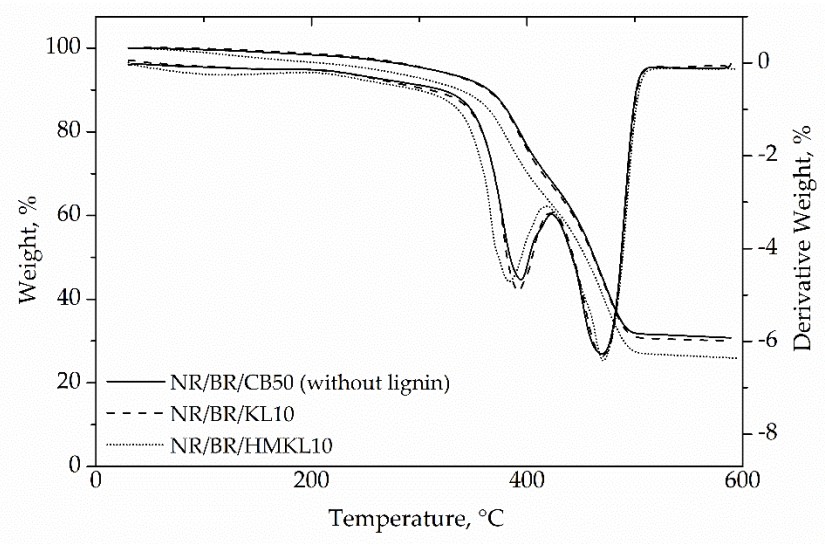

**Figure 15.** The effect of HMKL content on the thermal stability, thermogravimetric analysis (TGA), and derivative weight loss (DTG) of lignin-filled NR/BR compounds.

The thermogram curves in Figure 15 show the degradation of the rubber composites, which involved the degradation of the main chain of the rubber matrix via scission of the C–C bonds. Two steps of degradation can be observed in the thermogram curves. The weight loss of the rubber samples started at temperatures of around 180–200 °C. This is associated with volatile components, i.e., stearic acid, as well as the adsorption of water at around 300 °C. The first degradation started at 350 °C and terminated at approximately 400 °C. In this phase, the natural rubber segment (polyisoprene) was degraded, as indicated by the presence of a small peak in the DTG curve [66].

Natural rubber degradation is sensitive to the presence of an oxidized structure and the depletion of sulphidic crosslinks. However, the thermogram in Figure 15 shows that the DTG peak ($T_{max}$) for NR/BR/HMKL was shifted towards a lower temperature, i.e., from 392 °C to 383 °C for NR/BR/KL

and NR/BR/HMKL compounds, respectively. This shows there was a slight reduction in the thermal stability of the NR/BR composite. Calco-Flores et al. [66] reported that unmodified lignin is extremely effective at protecting rubber matrix from oxidation and high temperature due to the stabilizing effect of lignin's hindered phenolic hydroxyl groups towards reactions induced by oxygen and its radical species. However, the modification of lignin slightly minimized the oxidative degradation of the rubber. Additionally, it reduced the radical scavenging efficacy [40] of the attached hydroxymethylated groups at the free reactive site (C5) of the phenolic structure and secondary aliphatic groups.

The second phase of decomposition occurred between 450 and 550 °C. This phase (shown by a major peak in the DTG curve) was due to the cleavage of cross-linked BR and conjugated polyene left after the first phase of degradation [67]. During this phase, a similarly large DTG peak ($T_{max}$) is visible, which indicates that the thermal stability was slightly higher at high temperature. Meanwhile, there was more char residue from the compounds with KL than from the compounds containing HMKL. This shows that the compounds with the unmodified lignin were more stable than HMKL-filled compounds.

## 4. Conclusions

In this study, a detailed characterization of structural kraft lignin, both unmodified and modified by hydroxymethylation treatment, was accomplished. The results show that the hydroxymethylation treatment formed methylene bridges which bound together with the macromolecular lignin. It can be concluded that the inclusion of HMKL in rubber composites weakened the filler–filler interaction and improved the rubber's processability. Furthermore, the compatibility and high interfacial adhesion between the HMKL and the rubber matrix increased the cure rate of the rubber compound, and hence, increased the crosslink density compared to KL-filled NR/BR composites. A consequent improvement of the mechanical properties of the rubber composite at low HMKL contents of up to 10 phr was clearly observed. However, the aging resistance and thermal stability did not alter significantly for HMKL-filled NR/BRs as compared to the KL-filled NR/BRs. As a result, these properties influenced the flexing resistances of NR/BR composites containing HMKL, whose flexing resistances were lower than those of the NR/BR/KL composites. Replacing carbon black with HMKL at a content of up to 10 phr improves the rheological, curing, and mechanical performance of rubber composites. However, it slightly lowers their thermal stability and flexing resistance.

**Author Contributions:** Conceptualization, N.A.M.A. and N.O.; methodology, N.A.M.A., N.O., and M.H.H.; Project administration, N.O. and N.A.M.A.; formal analysis, N.A.M.A., N.O., K.S., and N.H.; investigation, N.A.M.A.; validation, N.O., K.S., N.H., and M.H.H.; resources, N.O., M.H.H., and K.S.; data curation, N.A.M.A., N.O., and K.S.; writing—review and editing, N.A.M.A., N.O., K.S., N.H., and M.H.H.; visualization, N.A.M.A.; supervision, N.O., K.S., and N.H.; funding acquisition, N.O. and M.H.H.

**Funding:** This research was funded by the Fundamental Research Grant Scheme, grant number 203.PBAHAN.6071350 and by USM Bridging Grants, grant number 304.PKIMIA.6316041.

**Acknowledgments:** The authors acknowledge the support of the School of Materials and Mineral Resources Engineering (SMMRE), Universiti Sains Malaysia (USM), and Prince of Songkla University, Pattani Campus (PSU). One of the authors, Nor Aniza Mohamad Aini (N.A.M.A.) is thankful to the Ministry of High Education (MoHE) of Malaysia for the MyPhD scholarship and Postgraduate Research Attachment (PGRA) Fund, Institute of Postgraduate Studies, USM.

**Conflicts of Interest:** The authors declare no conflict of interest.

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
