# Peer review of "Hydroxymethylation-Modified Lignin and Its Effectiveness as a Filler in Rubber Composites"

_processes, doi:10.3390/pr7050315_

Round 1

Reviewer 1 Report

Processes-481638

By Aini et al.

In this manuscript (ms), the authors report the hydroxymethylation of a kraft kenaf lignin and its subsequent use as filler in rubber. The manuscript contains many data on testing of the rubber indicative of a significant knowledge from the authors on rubber processing and evaluation. The modification of lignin, however, is not strongly demonstrated; a better choice of techniques could have been made; and NMR interpretation contains several wrong statements. The benefits of modification are not very obvious and several reverse statements can be found in the interpretation of testing. It is also regrettable that no real control (50 phr BC, without lignin) has been included in this study. Finally, this manuscript is poorly written with numerous badly constructed sentences and numerous grammatical mistakes.

This manuscript requires major, but achievable, revisions prior to becoming publishable in any journal. Below are suggestions to improve its quality. Note that language mistakes are completely ignored in this review.

For non-experts, full names should be provided at first appearance of acronyms in the text and in figure captions, e.g. NR/BR, phr,… Note that KL would be a more logical abbreviation than UL for the non-modified lignin.

Authors mentioned an increase of MW during hydroxymethylation (L 295) but GPC data in p2 contradict this statement.

Why is ammonium hydroxide added to KOH in the hydroxymethylation reaction?

G bands are not clearly shown in FT-IR spectra (Fig. 2) to support the text that claims their disappearance.

NMR interpretation needs to be revised. The authors seem to confuse aliphatic OH on a phenyl ring and aromatic OH. All hydroxymethyl groups should be primary alcohols, whether on the ring or on the side chain. Aliphatic OHs cannot be taken as a proof of reaction on the side chains. All peaks assignments and interpretation should therefore be corrected accordingly. The 1H data seem indeed to indicate an increase of aliphatic OH in HMKL compared to “UL”, indirectly shown by the acetyl groups. In line 222, peak between 1.5 and 2.49 ppm is claimed to be due to methylene groups. It is not correct: it corresponds to acetyl groups. Authors also mentioned a signal with high intensity appeared in 13C NMR in the region 90-59 ppm (l 236). No new signal is detectable in this region.

Viscosity seems lower with HMKL, not higher as stated in l 245.

Payne effect does not show significant improvement with modification when similar loadings are considered, in contrast to what is stated in l 266.

It is surprising that the data, ML and VC, given in Table 2, which are commented by the authors as indicative of same processes do not follow the exact same trends when increasing the content of HMKL.

Interpretation of hardness does not seem to make much sense.

All mechanical tests should have been compared to a rubber prepared similarly with 50 phr of black carbon and no lignin at all.

The authors speak a lot of compatibility between modified lignin and rubber. Explaining in more details the type of compatibility expected between lignin and rubber who be a good addition to this study. What type of compatibility are we talking about? Physical? Chemical? If this is physical compatibility, e.g. hydrophobicity, the introduction of an OH group should be detrimental. If it is chemical, please introduce a scheme showing the type of reaction occurring.

Some examples of hydroxymethylated lignins can be found in the literature for addition in rubber composites. Literature search should be made again and introduction revised accordingly.

Finally, the benefits of modification are really not obvious in this study and make the reader wonder whether modification of lignin is worthy to reach such contradictory effects depending on what parameter is considered.

Author Response

Dear Editor,

First of all, I would like to thank for all the fruitful comments and positive feedback from you. All the comments have been addressed accordingly.

Herewith, I enclosed the response to my manuscript entitle “Chemically modified lignin by hydroxymethylation and its effectiveness as filler in rubber compositeswhich given the number 481638.

All the responses to the comment as attached in the file

Reviewer 2 Report

The manuscript shows the incorporation of lignin in rubber composites, which was modified with formaldehyde and compared with an unmodified lignin used in rubber. Overall the curing and mechanical properties was improved using the modified lignin due to a higher crosslink density.

The synthesis/modification with formaldehyde and the condensation to marcomolecules was proven by FTIR and H-NMR/C-NMR measurements in a good manner. Also the testing (meachnical properties, curing, aging, TGA) on the rubber material supports the findings given by the authors. I think this manuscript will be interesting for the readership, but small improvements/corrections can be done.

The comparison of the testing was shown to a reference using a content of 10wt% of the unmodified lignin. Since the effects were shown for the modified lignin in a range between 5 to 20wt%. It could be nice to present the whole set of data for unmodified lignin also in this complete range for having a better comparison between the two kind of data sets. At least the figures 5, 6, 7, 9, 10, 12, should be changed so that the marker of the reference (UL10) appears at a lignin content of 10% and not at 0%. This makes these diagrams easier to be recognized by the reader.

line 57: Since not everybody might be familiar with the abbreviation NR/BR, it would be useful to introduce it

line 132: Flory Huggins parameter X (chi) should also appear as the greek letter in the text. It would be also nice to show, which value was used for the calculation of the network density and to give the reference for it.

figure 1: equation (a) the C=O bond of the formaldehyde molecule is a bit distorted

figure 5: error bars can be helpful showing the effect between UL10 and HMKL in a better way

line 445: Post-curinc of C=C bonds were named to be responsible to increase stiffness and that this might introduce a better mechanical property during the life-time at high temperature applications. I think it could be also possible that the brittleness is going to be increased and the damping of the rubber material can suffer from this. I can also imagine that side reactions of the aromatic moieties during thermal aging influence the properties of the rubber material.

Author Response

(The authors gave the same response as above.)

Reviewer 3 Report

The present article described the characterisation of hydroxymethylated Kraft lignin as filler in rubber composites. The title is well chosen, in accordance with the content of the article. The study was straightforward and the interpretation was clear and easy to understand. The conclusions are in agreement with the results. Most of the references are recent and relevant for the study.

Please consider some comments from my side:

1. The experiments are fairly well designed and executed, but there are lacks of statistical analysis used in the study.

2. Please describe more detail the relationship between the experimental results and hydroxymethylated Kraft lignin as filler in rubber composites.

Author Response

Dear Editor,

First of all, I would like to thank for all the fruitful comments and positive feedback from you. All the comments have been addressed accordingly.

Herewith, I enclosed the response to my manuscript entitle Chemically modified lignin by hydroxymethylation and its effectiveness as filler in rubber compositeswhich given the number 481638.

All the responses to the comment as attached in the file

Round 2

Reviewer 1 Report

Processes-481638-rev

By Aini et al.

Although most requested revisions related to chemistry have been done, no efforts have been provided by the authors to improve the writing of this manuscript. This manuscript cannot be published in its current form due to the numerous language imperfections and errors that impair significantly its quality.

Some errors are indicated directly on the manuscript (see attachment), but corrections are not exhaustive due to the amount of work it represented for reviewer.

Serious edition work is now necessary from the authors to allow publication of this manuscript.

Author Response

Cover Letter in Response to Reviewer’s Comments

Miss Susan Ji

Editor

Processes, MDPI.

Our sincere thanks to all the fruitful comments. Really appreciate this very much.

We have answered all the comments in our manuscript entitle “Chemically modified lignin by hydroxymethylation and its effectiveness as filler in rubber composites”, which has given the number 481638.

Herewith, we submit a revised version of the manuscript for full further consideration to be published in your reputable journal. All the comments have been revised accordingly.

We hope that you have satisfied with all the answers and could process the manuscript to further stage.

Sincerely

Assoc. Prof. Dr. Nadras Othman

Universiti Sains Malaysia

Round 3

Reviewer 1 Report

Processes-481638-rev2

By Aini et al.

Although some efforts have been provided by the authors to improve the writing of this manuscript, it still contains a non-negligible number of language imperfections and errors that need to be corrected.

Some corrections are suggested in attachment. This is a lengthy paper that would gain in being shortened and synthesized, may be by coupling various parameters to avoid such a high number of figures, that all look very similar.

Finally, I recommend that this paper is read and corrected by an English speaking person.

Author Response

Dear Editor,

Thank you for all the comments given to my manuscript.

We have done the English editing services as required by reviewer extensively. These are the related information from the MDPI English Editing team for our manuscript that you may crosscheck;

Title: Chemically Modified Lignin by Hydroxymethylation and Its Effectiveness 
as Filler in Rubber Composites
Length in words: 8574
English editing ID: English-9869
MDPI manuscript ID: processes-481638
Author: Nadras Othman
Author email: srnadras@usm.my
Editing cost: 276.48 CHF

Hope you could consider our manuscript to be published in your reputable journal.

Thank you.

Nadras
